# Spatial and Temporal Variation of $NO_2$ Vertical Column Densities (VCDs) over Poland: Comparison of the Sentinel-5P TROPOMI Observations and the GEM-AQ Model Simulations

**Marcin Kawka** [1,*], **Joanna Struzewska** [1,2] and **Jacek W. Kaminski** [1,3]

1   Institute of Environment Protection—National Research Institute, 01-692 Warsaw, Poland
2   Faculty of Building Services, Warsaw University of Technology, 00-653 Warsaw, Poland; joanna.struzewska@pw.edu.pl
3   Institute of Geophysics, Polish Academy of Sciences, 01-452 Warsaw, Poland; jkaminski@ecoforecast.net
*   Correspondence: marcin.kawka@ios.edu.pl

**Abstract:** The TROPOMI instrument aboard Sentinel-5P is a relatively new, high-resolution source of information about atmosphere composition. One of the primary atmospheric trace gases that we can observe is nitrogen dioxide. Thanks to TROPOMI capabilities (high resolution and short revisit time), one can describe regional and seasonal $NO_2$ concentration patterns. Thus far, such patterns have been analysed by either ground measurements (which have been limited to specific locations and only to the near-surface troposphere layer) or numerical models. This paper compares the TROPOMI and GEM-AQ derived vertical column densities (VCD) over Poland, focusing on large point sources. Although well established in atmospheric science, the GEM-AQ simulations are always based on emission data, which in the case of the energy sector were reported by stack operators. In addition, we checked how cloudy conditions influence TROPOMI results. Finally, we tried to link the $NO_2$ column number densities with surface concentration using boundary layer height as an additional explanatory variable. Our results showed a general underestimation of $NO_2$ tropospheric column number density by the GEM-AQ model (compared to the TROPOMI). However, for the locations of the most significant point sources, we noticed a systematic overestimation by the GEM-AQ model (excluding spring and summer months when TROPOMI presents larger $NO_2$ VCDs than GEM-AQ). For the winter months, we have found TROPOMI $NO_2$ VCD results highly dependent on the choice of qa_value threshold.

**Keywords:** air pollution; $NO_2$; Sentinel-5P; TROPOMI; GEM-AQ; Poland

## 1. Introduction

Nitrogen oxides ($NO_x = NO + NO_2$) play a significant role in tropospheric chemistry processes. As oxidiser precursors, they contribute to the tropospheric ozone formation process. Most of $NO_x$ emissions are released as a form of NO molecules, but they quickly convert to $NO_2$. Thus, tropospheric $NO_2$ is commonly used as a more chemically stable proxy for $NO_x$ emissions [1]. There are two major types of $NO_x$ emissions–from road traffic and from industrial emissions, both of which originate from high-temperature combustion. The former is located at the earth surface and distributed proportionally to the road network, the latter at stacks located at bigger industrial incineration plants. For the most significant industrial $NO_x$ sources in Poland, stack height is roughly within the range of 100–300 m. This paper focuses on large point emitters since they are an issue of great concern, and their environmental impact exceeds the local scale.

There are several methods for obtaining gridded $NO_2$ estimates on a larger than local scale. To name the most significant: chemical transport models (CTMs) or online chemical weather models, spatial interpolation of station-based measurements, empirical models (such as Landuse regression LUR or socioeconomic regression [2]), remote sensing

(satellite or, much less common, on an operational scale—aerial). Since each method has its intrinsic strengths and weaknesses, the synergistic use of multiple sources of information and data-driven methods (also known as data assimilation or data fusion methods) is also gaining increasing attention [3,4].

Within the satellite remote sensing of atmospheric pollutants, significant progress has been made in recent decades, starting from the first operational ultraviolet spectrometer, which was capable of delivering gridded data with a pixel size of $40 \times 320$ km$^2$ (Global Ozone Monitoring Instrument—GOME) in 1995 [5], followed by SCIAMACHY aboard Envisat (in 2002 [6]) and GOME-2 [7].

An undeniable advantage of progress in satellite remote sensing of tropospheric $NO_2$ concentrations is the growing archival record of past measurements on a global scale. This makes them a powerful tool for spatial and temporal trend analysis [8–10] for environmental policy evaluation, industry, and development assessment.

Applications of a satellite-derived $NO_2$ column data archive covered numerous aspects, such as an assessment of the effectiveness of abatement strategies in China [11–13], tracking effects of economic cycles [14–16], and short-term regulations for events such as the 2008 Beijing Olympic Games [17]. Thanks to high spatial resolution, satellite sensors play a vital role as an alternative (to bottom-up estimations) means for estimating industrial emissions from large power plants, e.g., India [18], Greece [19], and South Africa [20], as well as from big cities in Mexico [21], France [22], and the USA [23]. Due to high spatial resolution, TROPOMI revealed information on emitters that was previously considered as challenging to estimate, e.g., compressor stations in Siberia [24], shipping emissions in African harbours [25], and ONG exploration in Canada [26].

Many studies attempt to validate satellite-born TROPOMI $NO_2$ measurements using airborne [27,28] and ground-based [29–31] spectrometers. The general conclusion is that they tend to underestimate the $NO_2$ column number in highly polluted regions and overestimate it in regions with low $NO_2$ tropospheric column content.

This paper aims to assess to what extent satellite-borne TROPOMI $NO_2$ measurement can be used to evaluate the results of the operational chemical weather forecast model (GEM-AQ). We also check if TROPOMI results are valid under cloudy weather winter conditions within a temperate climate. Finally, we attempt to link a satellite-borne tropospheric column with the near-surface concentrations using boundary layer depth as an additional regression variable.

## 2. Data and Methods

In this study, we have used TROPOMI observations, the GEM-AQ model 24 h forecast from the operational run, and the observations from the national air quality monitoring network.

### 2.1. TROPOMI

TROPOMI, onboard Sentinel-5P satellite, is one of the most recently available instruments capable of monitoring $NO_2$ concentration in the atmospheric column. TROPOMI has a heritage to both the Ozone Monitoring Instrument (OMI) and the SCanning Imaging Absorption spectroMeter for Atmospheric CartograpHY (SCIAMACHY). The Sentinel-5P is intended to extend the data records of these missions and to be a preparatory mission for the Sentinel-5. Thus, resolution and revisit time should be at least at the same level as for OMI and SCIAMACHY. Sentinel-5P performs, on average, one full and two partial scans over our area of interest per day.

The concentration retrieval algorithm (DOMINO, developed by KNMI) is based on the $NO_2$ spectral properties in ultraviolet. It has previously been used for OMI [32], and with minor improvements, it has been adopted to TROPOMI data [33]. The retrieval algorithm uses several auxiliary atmospheric parameters within the processing, including the atmospheric mass factor (AMF).

To provide the necessary meteorological data, the profile shape from the TM5-MP model is used (run at $1 \times 1°$ resolution [34]). The surface albedo information is from a monthly OMI climatology (on a $0.5 \times 0.5°$ resolution). Finally, a vertical column density (VCD) is provided by the algorithm in units mol/m$^2$ with a spatial resolution of approximately $7 \times 3.5$ km$^2$ (approx. $5.5 \times 3.5$ km$^2$ after 6 August 2019 [35]), aggregated as a tropospheric, stratospheric, and total vertical column.

We used a level 2 product of TROPOMI (S5P_OFFL_L2__NO2), processed automatically by Copernicus Scientific data hub up to 5 days after sensing. Data were downloaded from the data hub using DHuSget 0.3.4—an automatic sentinel data retrieving script. Within the level 2 product of TROPOMI, a quality assurance flag qa_value is provided for each pixel. This normalised flag is to be used as a threshold for discarding poor-quality retrievals from useful ones. Most authors use the default threshold value of 0.75 [2,3,36,37]. However, this highly limits the number of retrievals in temperate climate due to intensive cloud cover, especially during the winter months. According to TROPOMI ATBD [38], the value of 0.75 is recommended and should remove clouds and scenes covered by snow, ice, and other problematic retrievals. However, the value of 0.5 is also proposed as still good enough for model-comparison studies. A lower threshold (thus a larger number of accepted retrievals) may be necessary if we still want to calculate monthly averages for the winter season. Discussion on the optimal value of qa_value will be given as examples in the results section. Therefore, we decided to perform further processing using not only 0.75 but also 0.5 and 0.7 thresholds as a potential compromise.

Pixels that fulfil the above qa_value threshold requirement are also used to create masking layers, which are later used to calculate model-based monthly average $NO_2$ column and model-based surface $NO_2$ concentration.

As the first processing step, TROPOMI data were regrided to GEM-AQ rectangular grid of size $300 \times 470$ and grid step 0.025°, using ESA Atmospheric Toolbox [39]. Secondly, regrided data were aggregated into monthly average raster. The term *monthly average*, although commonly used, may be a bit misleading in this context. Depending on location and time of the year, the monthly average may be an aggregate of 10 (in winter) to 40 (in the summer) cloud-free scans per pixel. TROPOMI $NO_2$ column concentration is a scalar value. However, it is produced with averaging kernel—an averaging vector which describes how sensitive the instrument was to $NO_2$ at a given time, altitude, and location. The same averaging kernel was applied for the tropospheric $NO_2$ column calculated from the model data to make the GEM-AQ results comparable.

### 2.2. The GEM-AQ Model

The GEM-AQ is a semi-Lagrangian chemical weather model in which air quality processes (chemistry and aerosols) and tropospheric chemistry are implemented online in the operational weather prediction model, the Global Environmental Multiscale (GEM) [40] model, which was developed at Environment Canada. The gas-phase chemistry mechanism used in the GEM-AQ model is based on a modified version of the Acid Deposition and Oxidants Model (ADOM) [41], where additional reaction in the free troposphere was included [42].

The GEM-AQ model instance, run at the Institute of Environment Protection (Poland), is an ensemble member in CAMS50 and hence undergoes evaluation against satellite observation in the scope of CAMS84. However, the model output requested for column calculations reaches only 5 km. For the sake of this paper, the entire troposphere was used.

An earlier study based on the comparison of the tropospheric $NO_2$ column with GEM-AQ satellite observations and SCIAMACHY observations addressed the spatial correlation with total $NO_x$ emission fluxes [43]. Since the TROPOMI instrument provides significantly better resolution than Envisat SCIAMACHY, it is now feasible to focus on particular categories of emissions. We chose to focus on significant industrial $NO_x$ sources because of the intensive contrast to the local $NO_2$ background.

Significant emission sources within the model are driven by emission data from the national emission inventory. These data are based on annual reporting obligations, which the facilities' owners fulfil. Annual emissions are transformed into monthly emission rates using the weighting factor from annual emission profiles. Emission profiles are assigned to so-called SNAP categories [44]. In the case of $NO_x$ emissions over Poland, the largest point emissions are assigned to SNAPs 1 (energy production from coal burning), 3 (nonenergy manufacturing industry, e.g., concrete or steel production), and 7 (road transport). Traffic emissions are considered to be uniform during the whole year, while SNAPs 1 and 3 are expected to follow a typical pattern of high in winter, low in summer (Figure 1).

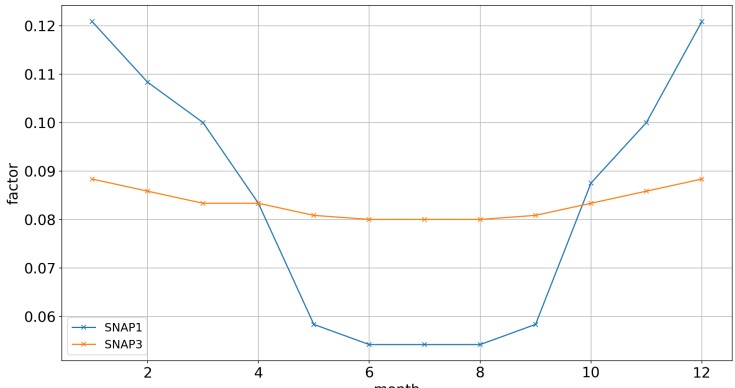

**Figure 1.** Annual emission profiles for the energy production sector (SNAP 1) and nonenergy manufacturing industry (SNAP 3).

The GEM-AQ model is set up to perform calculations using 28 vertical layers, out of which the lower 21 layers are classified as the troposphere. Troposphere averaging kernel is provided as an auxiliary variable of the TROPOMI level 2 $NO_2$ product. Averaging kernel values are provided at 35 levels of the TM5 model, which is the atmosphere model used within TROPOMI level 1 to level 2 processing [38]. TM5 averaging kernel is then linearly interpolated to GEM-AQ 28 levels (Figure 2). The $NO_2$ column number density is obtained for each layer using the following equation:

$$c_{NO_2,k} = f_k tnd_k \Delta z_k \tag{1}$$

where:

$f_k$ [ppb]—molecular mixing ratio
$tnd_k$ [molec/m$^3$]—total number density
$\Delta z_k$ [m]—layer depth

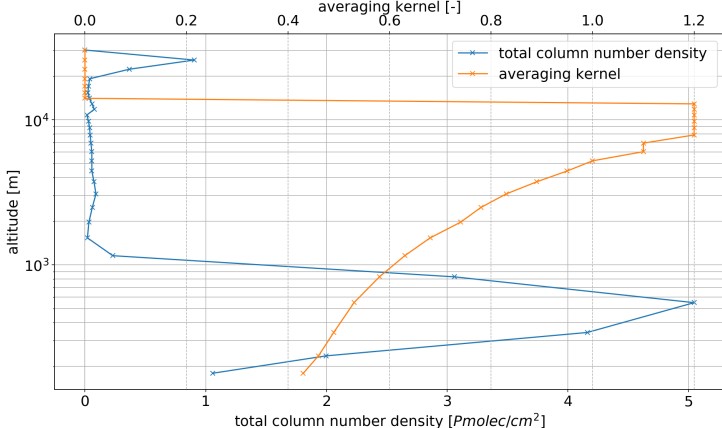

**Figure 2.** Example of $NO_2$ vertical profile from GEM-AQ model and the TROPOMI-derived, troposphere averaging kernel, extracted at power plant stack location on 1 April 2019.

The $NO_2$ column number density in the whole tropospheric column is then calculated using column number density from each GEM-AQ layer and averaging the kernel derived from TROPOMI image:

$$C_{NO_2} = \frac{1}{\sum_k avk_k} \sum_{k=1}^{28} c_{NO_2,k} avk_k \tag{2}$$

### 2.3. Boundary Layer Depth

The boundary layer is the lowest part of the troposphere, directly influenced by Earth's surface and responds to these forcings in a short time scale [45]. Significant $NO_x$ emissions occur within the boundary layer, while a satellite sensor observes the whole tropospheric column integrated. Therefore, we expect boundary layer depth to be an additional variable that explains to what extent the tropospheric column is affected by concentrations from the boundary layer.

There are several ways of estimating boundary layer depth. Since GEM-AQ is an online chemical weather model with the meteorological component, we decided to use Gradient Richardson Number $R_i$ with a critical value of $R_c = 0.025$. We assume that when $R_i < R_c$, we are within the boundary layer and turbulent mixing is the dominant form of transport [45]. The Gradient Richardson Number is calculated as

$$R_i = \frac{\frac{g}{\theta_v} \frac{\partial \theta_v}{\partial z}}{\left(\frac{\partial u}{\partial z}\right)^2 + \left(\frac{\partial v}{\partial z}\right)^2} \tag{3}$$

where $\theta_v$ is a virtual potential temperature, and $u$ and $v$ are horizontal components of the velocity vector, resulting from the meteorological part of the GEM-AQ model.

### 2.4. Surface Observations

Observations of surface $NO_2$ concentrations during 2019 were obtained from the Chief Inspectorate of Environment Protection, responsible for air quality monitoring in Poland. The dataset includes results from 112 automatic stations measuring with hourly time step.

## 3. Results

### 3.1. Overall Performance

Before detailed analysis, we performed a general linear regression analysis of the TROPOMI $NO_2$ tropospheric column retrieval. We expect tropospheric columns retrieved using the TROPOMI and the GEM-AQ models to be linearly correlated over the whole area of interest. Since we do not expect any additional bias, we assume that the noise is of Gaussian nature, and the following regression equation is expected to be fulfilled:

$$N_{v,GEM}^{trop} = a * N_{v,TROPOMI}^{trop} + b \tag{4}$$

where $N_{v,GEM}^{trop}$ is the monthly averaged GEM-AQ model-based tropospheric $NO_2$ column number density, $N_{v,TROPOMI}^{trop}$ is the monthly averaged TROPOMI-based tropospheric $NO_2$ column number density, and $a$ and $b$ are regression parameters.

Table 1 summarises fitting results. The best (in terms of high $R^2$ and low MSE) linear regression was obtained for the July monthly average tropospheric column. The MSE value follows the pattern of being low during summer months and higher during winter months. $R^2$ does not seem to reveal any annual pattern. Thus, it is either cloud cover or emission underestimation, making the GEM-AQ and TROPOMI tropospheric columns slightly different.

Both scatter plots (Figure 3) and regression parameters ($a < 1$) suggest that except for winter months, on a regional scale, the GEM-AQ model underestimates the $NO_2$

tropospheric column number densities. However, it is still questionable at this stage if it is an overestimation caused by TROPOMI or an underestimation by GEM-AQ.

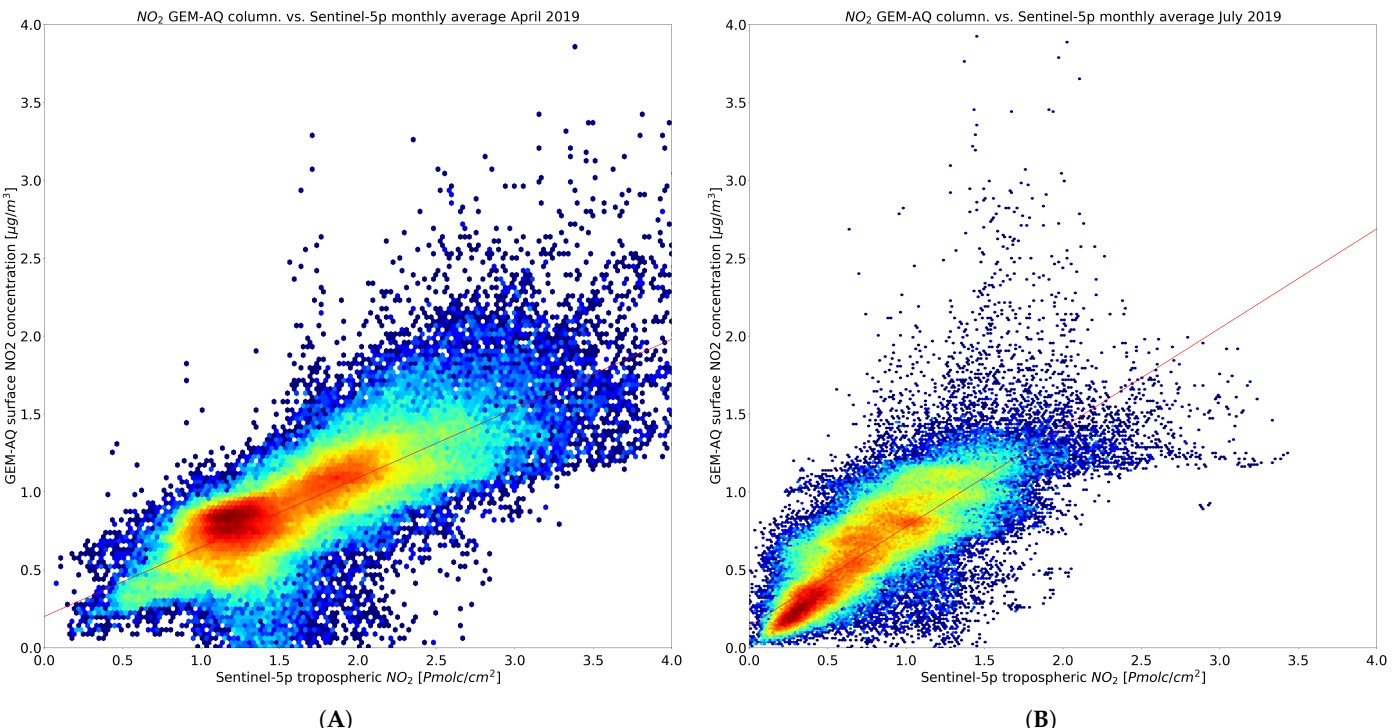

**Figure 3.** Scatter plots of monthly averaged tropospheric column retrieved by TROPOMI (horizontal axis) and GEM-AQ model (vertical axis) for (**A**) April and (**B**) July.

**Table 1.** Parameters of Equation (4), fitted to monthly averaged tropospheric column rasters; goodness of fit for each monthly average.

| Month | a | b | $R^2$ | $MSE$ |
|---|---|---|---|---|
| January | 1.4 | 0.28 | 0.13 | 1.29 |
| February | 0.92 | 0.3 | 0.36 | 0.67 |
| March | 0.38 | 0.44 | 0.36 | 0.11 |
| April | 0.45 | 0.2 | 0.53 | 0.05 |
| May | 0.42 | 0.23 | 0.59 | 0.06 |
| June | 0.51 | 0.22 | 0.37 | 0.04 |
| July | 0.64 | 0.14 | 0.66 | 0.04 |
| August | 0.59 | 0.15 | 0.45 | 0.04 |
| September | 0.54 | 0.29 | 0.52 | 0.06 |
| October | 0.75 | 0.06 | 0.63 | 0.14 |
| November | 0.78 | 0.09 | 0.5 | 0.48 |
| December | 0.8 | 0.58 | 0.4 | 0.69 |

### 3.2. The Choice of qa_value

TROPOMI $NO_2$ OFFL product was processed by the DOMINO algorithm (version 1.2) on the ESA side. One of the auxiliary outputs of this algorithm is the quality assurance flag (qa_value). According to TROPOMI $NO_2$ ATBD [38], the threshold of 0.75 should be used to remove clouds, pixels covered by snow, and other problematic retrievals. Setting the threshold at 0.75 is sufficient for summer months; however, in winter (November–February), only a few (less than 10) satellite images per month satisfy this condition (Figure 4C,D). Reducing qa_value threshold to 0.7 or 0.5 may lead to some improvement (Figure 4A,B). However, a lower threshold leads to an underestimation in comparison to modelling results (Figure 8).

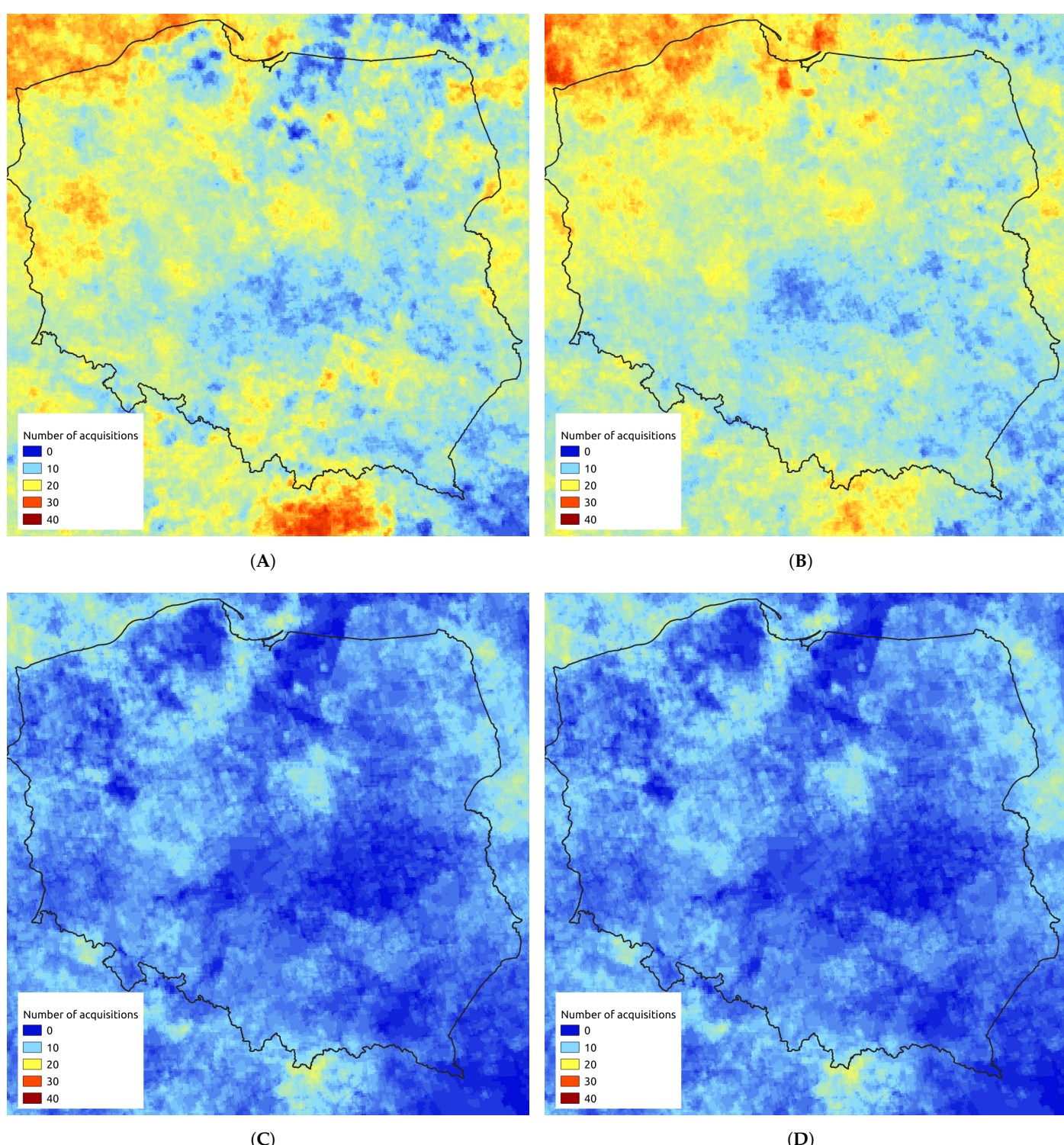

**Figure 4.** Number of pixels available for monthly averaging in January 2019 for given qa_value threshold (**A**) 0.5; (**B**) 0.7; (**C**) 0.75; (**D**) 0.8.

### 3.3. Spatial Distribution

We investigated the spatial distribution of $NO_2$. As Figure 5 shows, the monthly averaged TROPOMI tropospheric $NO_2$ column reproduces the locations of significant $NO_x$ point sources. At the same time, this spatial pattern is not reproduced in the GEM-AQ surface layer. This confirms the fact that the TROPOMI instrument at the satellite level is not sensitive to surface layers. The only location where the TROPOMI tropospheric column seems to be better correlated with surface concentration than with the GEM-AQ

tropospheric column is the coastal area, near the city of Gdańsk. Although no significant $NO_x$ point sources are located there, relatively high values of $NO_2$ column number density on the TROPOMI column (Figure 5A) and model-based $NO_2$ surface concentration (Figure 6B) can be noticed near the city of Gdańsk. This fact may be explained by harbour emissions, which may be underestimated in the GEM-AQ model. Due to local sea breeze circulation, the whole tropospheric column could be well mixed.

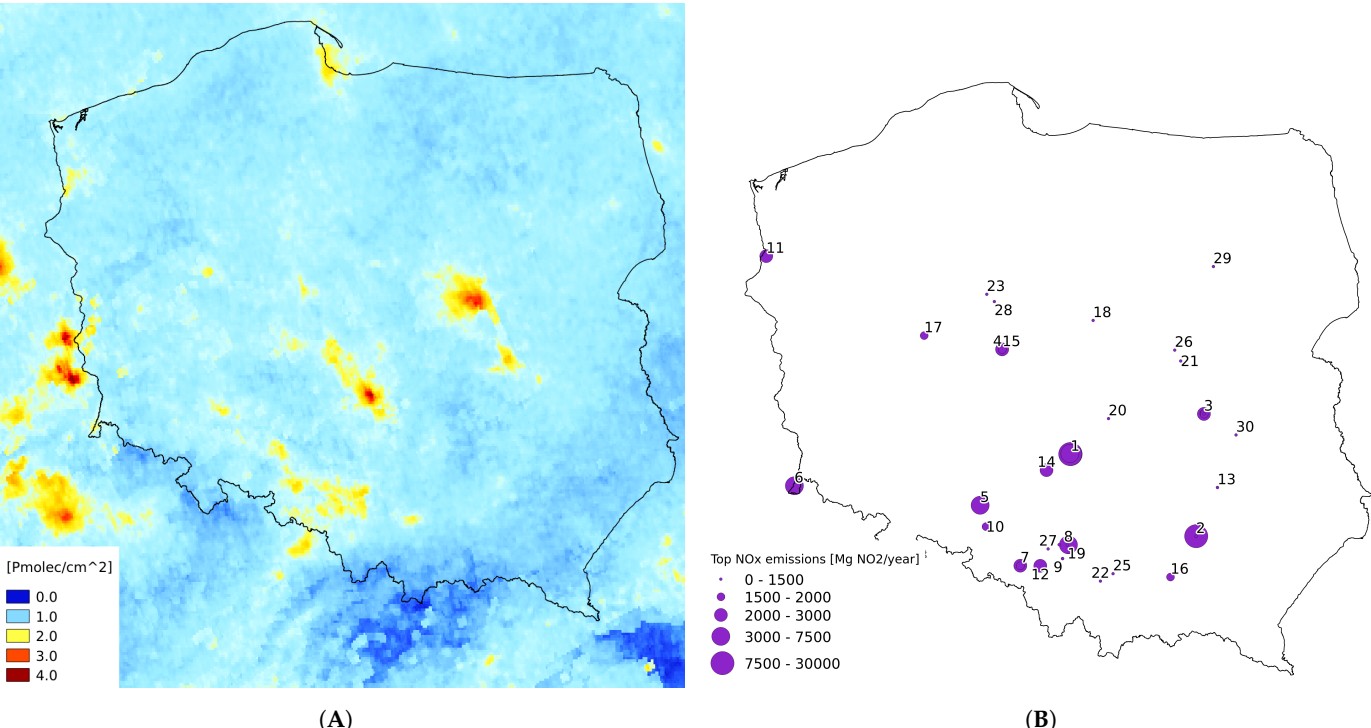

(**A**)　　　　　　　　　　　　　　　　　　　　(**B**)

**Figure 5.** (**A**) Satellite distribution of $NO_2$ tropospheric column density (monthly average—June 2019); (**B**) Locations of top 30 $NO_x$ emission point sources in Poland.

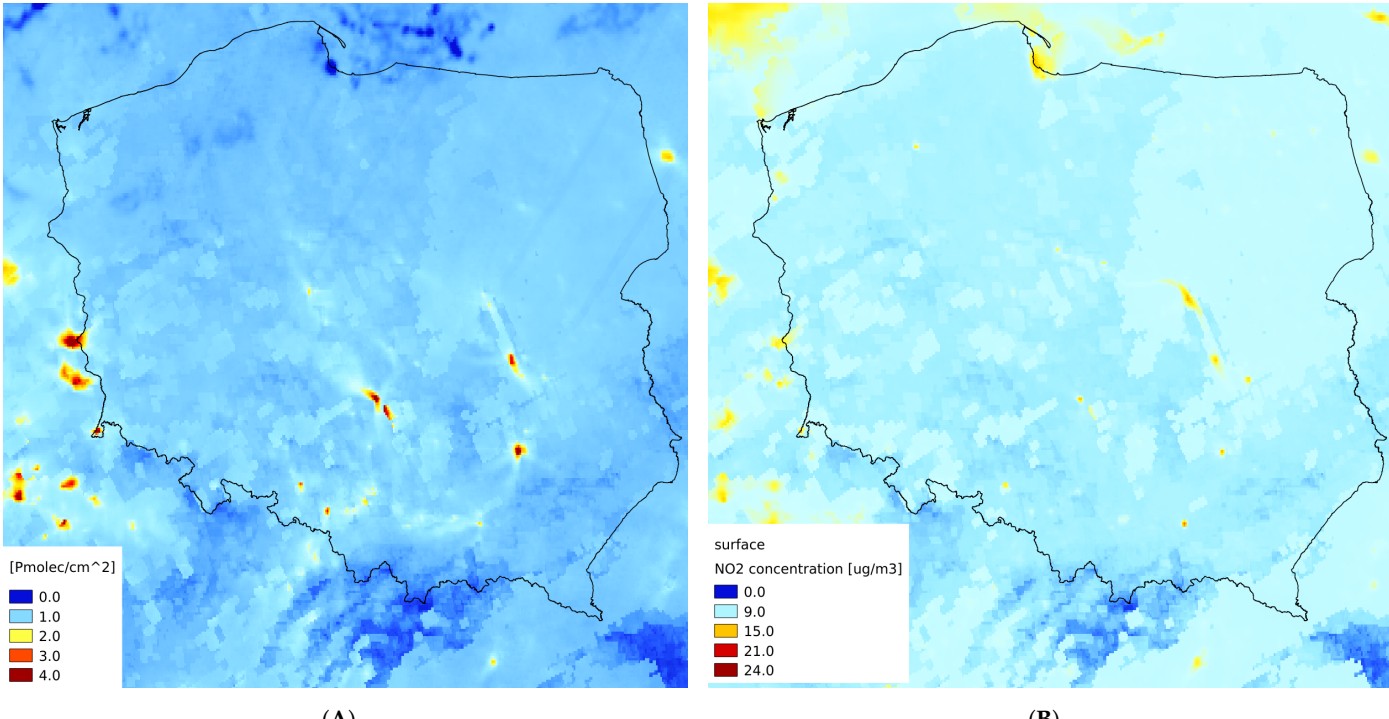

(**A**)　　　　　　　　　　　　　　　　　　　　(**B**)

**Figure 6.** (**A**) GEM-AQ model-based $NO_2$ tropospheric column density (monthly average—June 2019); (**B**) GEM-AQ model-based $NO_2$ concentration at surface level (monthly average—June 2019).

The troposphere $NO_2$ column number density reveals the locations of primary point sources and the dominant wind direction. As the monthly average tropospheric $NO_2$ column is an average of noncloudy days (mornings), the resulting spatial distribution depends on accidental wind direction. A comparison of the model-based and satellite-borne tropospheric $NO_2$ column over the whole domain (such as Figure 3 and Equation (4)) may be biased by a small error in modelled wind direction, leading to the wrong concentration distribution.

To make the comparison less wind-dependent, we extracted the troposphere column number density value from pixels surrounding the locations of the fifteen major point emitters (Figure 5B) from both TROPOMI results and the GEM-AQ model. Yearly-averaged values for GEM-AQ and TROPOMI in most cases agree within the margin of 15%. Only in the case of three emitters (out of fifteen) does the GEM-AQ model seem to underestimate the tropospheric column number density; in the other cases, a slight overestimation by the model is visible (Figure 7).

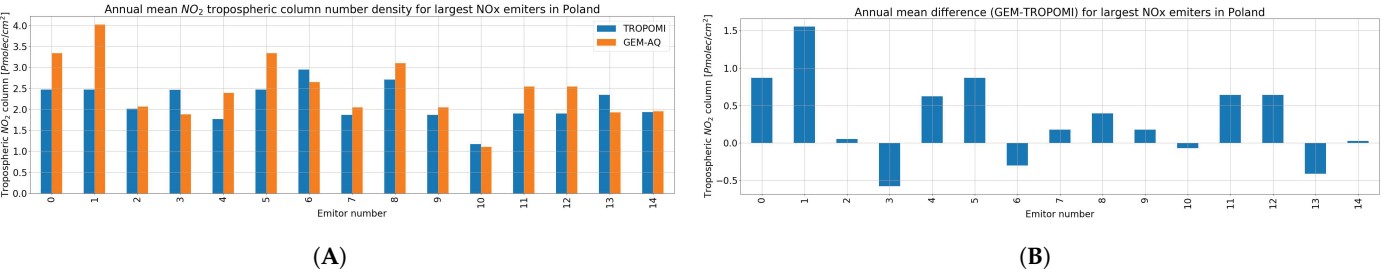

**Figure 7.** (**A**) Annual mean tropospheric $NO_2$ column number density extracted from pixels surrounding fifteen largest point emitters, from both TROPOMI acquisition and the GEM-AQ model. (**B**) Annual mean difference between GEM-AQ and TROPOMI tropospheric column for fifteen largest emitters; emitter numbers are the same as in Figure 5.

*3.4. Temporal Comparison*

Both $NO_x$ emissions and $NO_2$ concentrations follow the same annual pattern—low in summer and high in winter. This fact is due to higher energy demand and low wind velocity episodes during the winter months. Moreover, some of the largest coal-burning power plants in Poland are also sources of heat for city-wide heating systems. Thus, they burn more coal during low temperature periods.

Because of cloud cover and nonpoint sources (road traffic), we decided to analyse the temporal pattern only over the largest $NO_x$ emitters. The difference between the GEM-AQ tropospheric column and TROPOMI tropospheric column seems to be the smallest during the summer months (less than 0.5 Pmolec/cm$^2$, Figure 8). In autumn, the difference starts to grow, and it exceeds 1.0 Pmolec/cm$^2$ in December.

The choice of the qa_value threshold seems to have a significant influence in January and February. For qa_value = 0.5, TROPOMI returns higher values than the GEM-AQ model. This is probably due to partial cloud cover, which would have been filtered out when a higher qa_value is chosen. Regardless of the qa_value, April and May $NO_2$ column concentrations seem to be underestimated in the GEM-AQ model, which may be caused by an overestimated ozone production during these months.

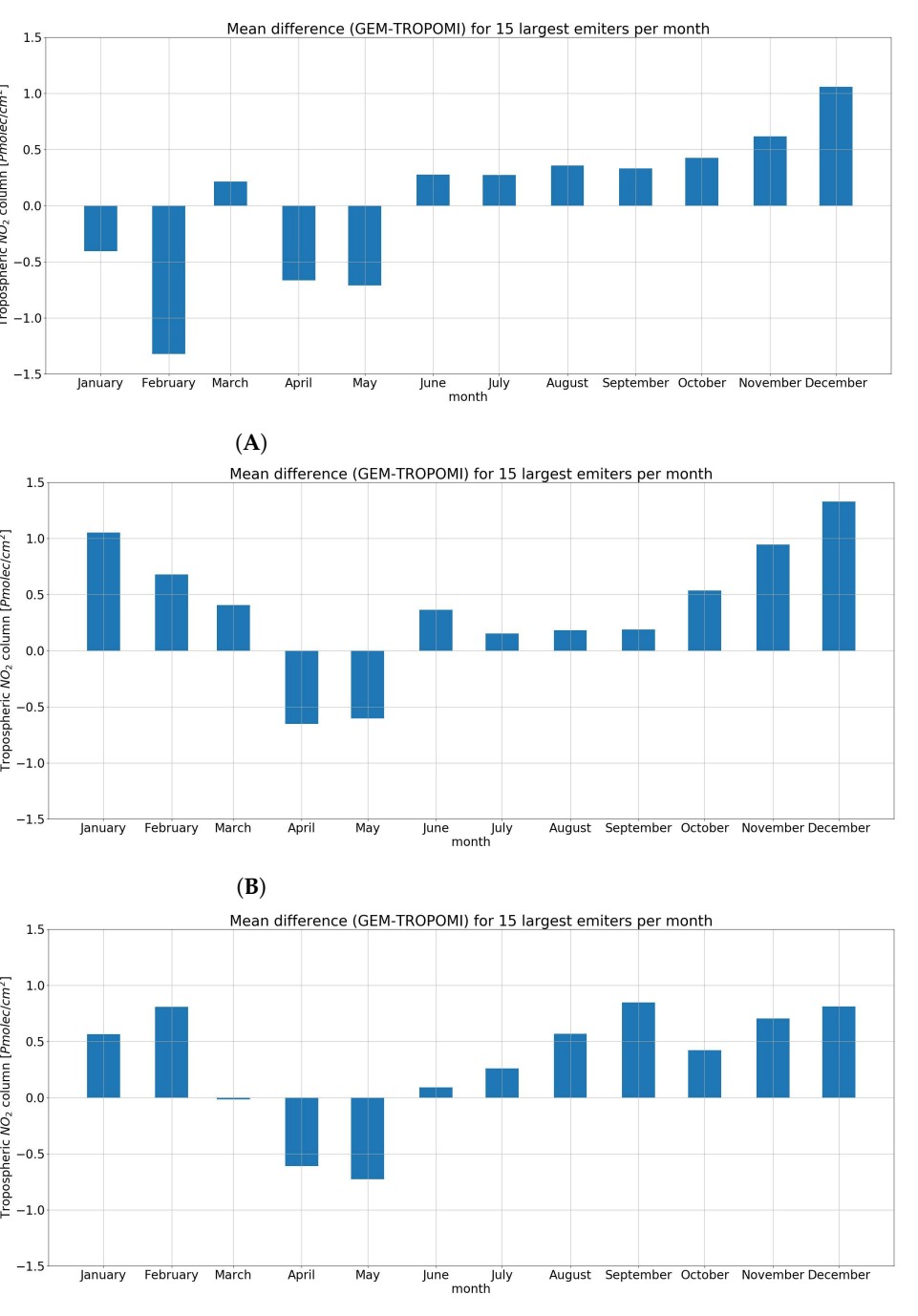

**Figure 8.** Mean difference in tropospheric column number density per month for different qa_value threshold: (**A**) *qa_value* = 0.5, (**B**) *qa_value* = 0.7, (**C**) *qa_value* = 0.75.

### 3.5. Relation to Near-Surface Concentration

Although there are authors [2] who present a linear relation between the near-surface concentration and $NO_2$ tropospheric column, according to the averaging kernel vertical distribution, the TROPOMI instrument is not very sensitive to $NO_2$ concentrations at surface level (Figure 2). It is probably hindered by the sensitivity at higher levels of the troposphere. Therefore, a more complex relation linking the $NO_2$ near-surface concentration and the tropospheric column is needed.

A concept of explaining tropospheric $NO_2$ column density using nonlinear regression against surface concentration and boundary layer depth was introduced by Dieudonne et al. [46]. Later on, it was applied to TROPOMI data over Paris by Lorrente [22], who

showed $NO_2$ surface concentration $c_{surf}$, tropospheric vertical column number density $N_v^{trop}$, and boundary layer depth $h$ in the following empirical equation:

$$N_v^{trop} = K[0.244h(c_{surf} - 1.38) + 0.184(c_{surf} - 2.83)] \quad (5)$$

where $K$ is a constant conversion factor ($1.31 \cdot 10^{15}$ molc/cm$^2$). We decided to introduce a more general nonlinear equation:

$$N_v^{trop} = [(a \cdot h + b) \cdot c_{surf} - c \cdot h + d] \quad (6)$$

The parameters of Equation (6) were fitted using the Levenberg–Marquardt algorithm [47]. Fitting was performed separately for each measurement station in each month. Stations where the number of TROPOMI tropospheric column values were lower than ten were discarded. Therefore, the results were highly dependent on cloud conditions. The best results were obtained for April and September 2019 (Figure 9). The spatial pattern of the correlation coefficient reveals that Equation (6) performs reasonably well within larger cities and densely populated areas (Figure 9). This is probably caused by the more significant contribution of road traffic $NO_x$ emissions to the tropospheric $NO_2$ VCD.

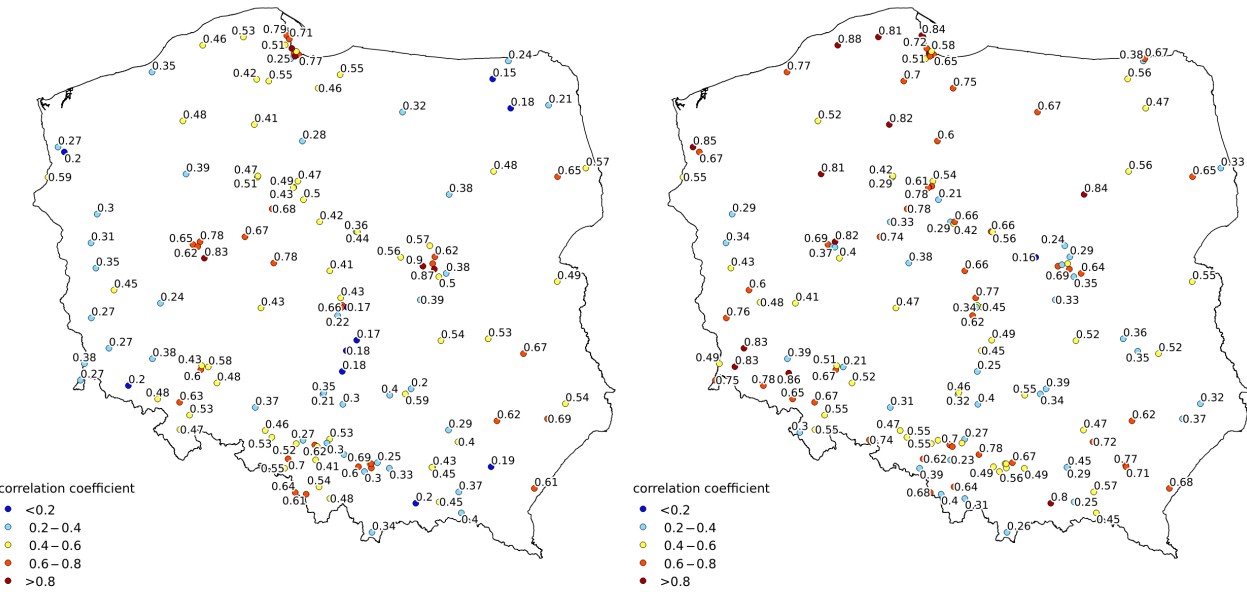

**Figure 9.** The correlation coefficient for nonlinear regression Equation (6) in (**A**) April 2019 and (**B**) September 2019.

## 4. Conclusions

Over the operational air quality forecast domain, performed routinely using the GEM-AQ model, we examined the results of the latest fine-scale satellite instrument (TROPOMI aboard Sentinel-5P) from the year 2019. The key findings from this study are the following:

1.  In general, the GEM-AQ model tends to underestimate the $NO_2$ tropospheric column number density, which may be caused by either too intense of mixing in the atmosphere, a sink of $NO_2$ into further chemical processes (e.g., tropospheric ozone production), or too small of a background concentration.
2.  When looking at locations next to the largest $NO_x$ point emitters in Poland, the GEM-AQ model and TROPOMI converge reasonably well. Minor differences should be explained by individual emission examination.
3.  The TROPOMI instrument does not correctly reproduce the annual temporal concentration pattern. It seems that cloud cover (thus qa_value threshold) and the number of satellite scenes averaged into a monthly average play an important role. Lowering the

qa_value during the summer months improves the convergence between TROPOMI and GEM-AQ, while during the winter months, it acts oppositely.

4.  The relation between near-surface concentration and troposphere column number density can be parametrised using boundary layer depth as an additional explanatory variable.

We conclude that TROPOMI is powerful and independent from the source of ground measurements of the $NO_2$ distribution data from the above findings. Although column number density is not to be used directly with surface concentration, it is still helpful for validating modelling results. After some additional processing, TROPOMI $NO_2$ column number densities can also be used for estimating near-surface concentrations in urban areas.

In further works, we would like to broaden our studies to model runs with different emission inventories–CAMS and EMEP. An interesting follow-up study would also be developing a data assimilation scheme for the GEM-AQ model capable of assimilating TROPOMI $NO_2$ VCDs into the GEM-AQ model.

**Author Contributions:** Conceptualization: J.S.; methodology, software, validation, resources, data curation: M.K.; writing—original draft preparation: M.K.; writing—review and editing: J.S.; visualization: M.K.; supervision: J.W.K.; project administration, J.W.K. All authors have read and agreed to the published version of the manuscript.

**Funding:** This research received no external funding.

**Institutional Review Board Statement:** Not applicable.

**Informed Consent Statement:** Not applicable.

**Data Availability Statement:** Sentinel-5P data are free from Sentinel-5P Datahub. GEM-AQ model results belong to operational air quality forecast, which is run at the IEP-NRI, Poland—a subcontractor of the Polish Chief Inspectorate of Environment. As a state-ordered calculation—results are publicly available on request.

**Conflicts of Interest:** The authors declare no conflict of interest.

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
