# Peer review of "Spatial and Temporal Variation of NO2 Vertical Column Densities (VCDs) over Poland: Comparison of the Sentinel-5P TROPOMI Observations and the GEM-AQ Model Simulations"

_atmosphere, doi:10.3390/atmos12070896_

Round 1
Reviewer 1 Report
This is an interesting, clearly written paper. I recommend it be published after addressing the following minor comments.
line 33: what is "res. XX"?
line 73: "Exact the number" is unclear.
line 110: "Traffic emissions are considered to be uniform during the whole year" - what is the impact of this assumption? Certainly different times of year have different traffic volume, for example, periods when children are in school vs. their summer vacation.
Author Response
We thank the Reviewer for the careful and insightful review of our manuscript. We address all of the concerns of the referee below:
line 33: what is "res. XX"?
Response: The "res. XX" was an "orphan" from previous editions and has been removed.
line 73: "Exact the number" is unclear.
Response: This sentence has been restructured. It refers to the optimal choice of qa_value. The influence of selecting different qa_value thresholds is presented in the Results section.
line 110: "Traffic emissions are considered to be uniform during the whole year" - what is the impact of this assumption? Certainly different times of year have different traffic volume, for example, periods when children are in school vs. their summer vacation.
Response: We agree that uniform in time traffic NOx emissions is a vast simplification. However, in this paper, we focus on industrial emissions as their influence is easier to estimate for the following reasons:
- the magnitude of industrial emission in Poland is more significant than the transport one; thus, the traffic emission is hindered by the industrial one at stack location.
- stack heights of the most significant polish industrial emitters are between 120 and 300 meters. Moreover, as the temperature of the fumes is often much higher than the ambient air, they are uplifted to the upper troposphere. This makes them relatively more "visible" to the sensor at satellite level (see fig 2. with averaging kernel from the paper) than the car exhaust fumes at the near-surface level.
- the stack locations are well known, and the emission magnitude does not vary within several hours.
Reviewer 2 Report
This paper is interesting and shows promise but I recommend major revisions to address the following points:
- Improved literature review.
- Improved abstract and results summary.
- Improved english and figure presentation.
- Improved summary and recommendations for future work
- Improved literature review and TROPOMI NO2 validation studies.
There have been many recent studies both evaluating TROPOMI NO2 as well as using TROPOMI NO2 for pollution variability. The literature review is insufficient in my opinion and should reference some selected recent studies and place this work in the context of these studies. What does this study contribute that is novel in terms adds to the knowledge base on using NO2 to look at cities and evaluating models. I believe the novel information is here, but not pointed out and the literature review is too weak.
Please see this paper for examples and references of both recent TROPOMI NO2 literature and validation: Remote Sensing | Free Full-Text | Meteorological Drivers of Permian Basin Methane Anomalies Derived from TROPOMI (mdpi.com)
I have copied the relevant text from the linked paper below:
“The high spatial resolution of TROPOMI measurements has led to recent studies investigating the impacts of meteorological, social, or economic forcing (e.g., COVID-19 pandemic) on the spatial and temporal variability of NO2 [26,27,28,29,30,31]. Recent studies across the world (e.g., Europe [32,33], Canada [34], USA [35], and China [36]) have linked changes in NO2 to meteorological variations (e.g., solar angle and wind speed). In Alberta, Canada, TROPOMI NO2 observations were validated over plumes from individual mining facilities [37]”…
“TROPOMI tropospheric and total NO2 column data have been compared against NO2 spectrometers, with a general negative bias (−23% to −51%) for tropospheric column data attributed to errors in chemical transport models, cloud effects and aerosols [40]. Other studies, comparing aircraft- and ground-based spectrometer measurements in urban and non-urban areas. have also found a systematic underestimation in TROPOMI NO2 measurements, particularly during highly polluted conditions [34,41,42,43,44,45].”…
- Improved abstract and results summary and future work
The abstract does not describe the study results, only what was done in the study:
What was done: In this paper, we attempted to 7 compare the TROPOMI and GEM-AQ derived VCDs over Poland with a particular focus on large 8 point emitters. We also checked how cloudy conditions influence TROPOMI results. Finally, we tried 9 to link the NO2 column number densities with surface concentration using boundary layer height as 10 an additional explanatory variable.
Please add what your results were after this in the abstract.
Please also add a paragraph of discussion describing the importance of this study and putting it in the context of other work and suggesting any future work at the end of the paper after the bullet points are provided.
- Improved english and figure presentation.
Please give a thorough English edit throughout. There are numerous sentences that are awkward and some spelling errors:
Examples, TROPOMI spelled TRPOMI in abstract.
Line 3: “By now, we were using” is a poor choice of words. Should be replaced with something better.
Line 17: Please add some clarifying modifiers to this sentence (something along the lines of what I wrote in caps). “There are two major types of NOx emissions – FROM ONROAD traffic and FROM industrial emissions, BOTH OF WHICH originate from high temperature combustion.
Please also make sure the font on the figures can be seen. They are too small in my opinion in figure 1 and 2.
Author Response
We thank the Reviewer for the careful and insightful review of our manuscript. We address all of the concerns of the referee below:
Improved literature review.
Response: The literature review has been broadened. We believe that the Introduction section introduces the state of the art of TROPOMI applications from recent papers.
Improved abstract and results summary and future work
The abstract does not describe the study results, only what was done in the study:
Please add what your results were after this in the abstract.
Response: The abstract has been rewritten, including results description
Please also add a paragraph of discussion describing the importance of this study and putting it in the context of other work and suggesting any future work at the end of the paper after the bullet points are provided.
Response: We have added a paragraph describing further works at the end of the Conclusion section.
Improved english and figure presentation.
Please give a thorough English edit throughout. There are numerous sentences that are awkward and some spelling errors.
Response: Spelling errors pointed out by the reviewers, and several additional ones have been fixed.
Please also make sure the font on the figures can be seen. They are too small in my opinion in figure 1 and 2.
Response: Figures 1 and 2 have been fixed.
Reviewer 3 Report
The subject analyzed in this article is very interesting and valuable. The article should be published in its current form with some minor modifications regarding spelling errors, such as TROPOMI in the abstract, lines 73-74, line 167 three instead of tree...
Moreover, including additional statistical parameters could strengthen the studies carried out.
Author Response
The subject analyzed in this article is very interesting and valuable. The article should be published in its current form with some minor modifications regarding spelling errors, such as TROPOMI in the abstract, lines 73-74, line 167 three instead of tree…
Response: Thank you for pointing out the spelling errors. We have corrected them.
Moreover, including additional statistical parameters could strengthen the studies carried out.
Response: It is a bit unclear what kind of statistical parameters Reviewer meant. So far, we are analysing the following statistical parameters:
- R2 and MSE for linear regression between monthly average NO2 tropospheric column (from TROPOMI and GEM-AQ) – Table 1. in the submitted manuscript
- Annual mean difference (MBE) for largest NOx emitters in Poland (fig. 7 B)
- Monthly mean difference (MBE) for largest NOx emitters in Poland, for different qa_value (fig. 8)
- The correlation coefficient for multi-parameter regression (Eq. 5), which describes NO2 tropospheric column number density as a function of boundary layer depth and near-surface NO2 concentration
Reviewer 4 Report
We read article and we have question.
Page 4
What is "Gradient Ricardson"? Maybe it "Gradient Richardson number"?
We do not understand what is "qa_value"?
Why are some lines not numbered?
The authors write in the conclusion: "From the above findings, we conclude TROPOMI is a powerful and independent from ground measurements source of NO2 distribution data. Although column number density is not to be used directly with surface concentration, it is still useful for validating modelling results, and after some additional processing, it can also be used for estimating surface concentrations in urban areas." In this case, the limitations of TROPOMI were previously indicated. In this regard, it can be considered a TROPOMI powerful source of NO2 distribution data? Does it make sense to use this instrument for research, including research options those shown in the conclusion?
Author Response
We thank the Reviewer for the careful and insightful review of our manuscript. We address all of the concerns of the referee below:
Page 4: What is "Gradient Ricardson"? Maybe it "Gradient Richardson number"?
Response: It's Gradient Richardson number. Thank you for pointing it out.
We do not understand what is "qa_value"?
Response: qa_value is an additional spatial variable provided by the processing algorithm with TROPOMI level 3 data (tropospheric column number density) in the same netCDF file. It is a continuous quality descriptor, varying between 0 (no data) and 1 (full quality data). It is recommended to ignore NO2 column number density data from pixels with qa_value < 0.5. Pixels with qa_value between 0.5 and 0.75 should be interpreted with care as their values may be biased by incidental factors like cloud cover.
Why are some lines not numbered?
Response: Paragraphs with equations had unnumbered lines. We have fixed this by putting each equation in a separate paragraph.
The authors write in the conclusion: "From the above findings, we conclude TROPOMI is a powerful and independent from ground measurements source of NO2 distribution data. Although column number density is not to be used directly with surface concentration, it is still useful for validating modelling results, and after some additional processing, it can also be used for estimating surface concentrations in urban areas." In this case, the limitations of TROPOMI were previously indicated. In this regard, it can be considered a TROPOMI powerful source of NO2 distribution data? Does it make sense to use this instrument for research, including research options those shown in the conclusion?
Response: To our best knowledge, TROPOMI brings added value to research due to the following reasons:
- it captures the whole tropospheric column, so it's useful for additional validation of the results of air quality models (in our case, chemical weather model) that provide results in 3D. This means validating both spatial and temporal distribution.
- Although there were instruments like GOME or SCIAMACHY capable of delivering NO2 column number density, the resolution of TROPOMI is much better. It makes it easier for researchers to focus on particular point sources.
Round 2
Reviewer 2 Report
The authors did a great job with improved literature review and addressed all my concerns, so I support publication. I would change the wording of "further works" to future work (minor wording suggestion) at the end.